# The Influence of Arabinoxylan of Different Molar Masses on the Properties of Rye Bread Baked by the Postponed Baking Method

**DOI:** 10.3390/foods13162482

**Published:** 2024-08-07

**Authors:** Angelika Bieniek, Krzysztof Buksa

**Affiliations:** Department of Carbohydrate Technology and Cereal Processing, University of Agriculture in Krakow, Balicka 122, 30-149 Krakow, Poland; angelika.bieniek@student.urk.edu.pl

**Keywords:** rye flour, arabinoxylan, postponed baking, rye bread, cross-linked arabinoxylan

## Abstract

Rye grain is a good source of dietary fiber, phenolic compounds, vitamins, and mineral compounds. To prevent the staling process of bread, semi-finished bakery products are subjected to cooling or freezing, and this process is called the postponed baking method. The aim of this study was to examine the influence of rye arabinoxylans differing in molar mass on the properties of rye bread baked using the postponed baking method. The breads were baked from rye flour types 720 and 1150, without and with a 1% share of unmodified or cross-linked rye arabinoxylans (AXs). The molar mass of the unmodified AXs was 432,160 g/mol, while that of the AXs after cross-linking was 1,158,980 g/mol. The results of this study show that the 1% share of AXs significantly increased the water addition to both types of rye flour and dough yield, and this increase was proportional to the molar mass of the AXs used. It is shown that a 1% share of both AX preparations positively increased the volume and crumb moisture of bread baked by the postponed baking method. Cross-linked AXs proved to be particularly effective in increasing the volume and bread crumb moisture. Both AX preparations had a positive effect on reducing the bread crumb hardness of rye breads baked by the postponed baking method.

## 1. Introduction

Rye grain is used to produce bread, especially in Northern and Central Europe [1]. Rye grain, commonly consumed in whole-grain products, is a good source of dietary fiber, phenolic compounds, vitamins and minerals [1]. Additionally, it has been shown that consuming rye grain products has a positive effect on glucose metabolism and the feeling of satiety in the human body [1].

Polysaccharides such as starch and arabinoxylans, as well as enzymes that degrade these polysaccharides, are particularly important when preparing dough for rye bread [2,3,4]. Proteins are less important in forming the structure of rye dough compared to wheat bread [3,4].

Arabinoxylans (AXs) are non-starch polysaccharides and their content in rye grain is 2.2–12.2% [5,6,7,8,9]. The molar mass of natural AXs from rye grain is 197,800–2,000,000 g/mol [6,7,9,10,11]. The water-soluble (WEAX) and water-insoluble (WUAX) fractions of arabinoxylans have specific functions, especially in rye dough and bread. Arabinoxylans can be isolated from rye grain and then added to dough to increase their levels. Nevertheless, the improvement of bread quality, defined as high bread volume, as well as low hardness and high moisture of the bread crumb, largely depends not only on the type and quantity, but also on the properties of AX preparations obtained by isolation [7,12,13,14,15]. To obtain better and more predictable effects of using AXs, they can be modified by methods used in food technology, such as cross-linking, hydrolysis, or fractionation [7].

While studying the effect of added preparations of water-soluble AXs (WEAX) isolated from rye and wheat grains, as well as water-insoluble wheat AXs (WUAX) on the properties of wheat doughs, an increase in water absorption, development time, and dough mixing has been observed. The addition of WEAX improved bread properties such as bread volume and the moisture and texture of the bread crumb. However, to achieve these effects, it is necessary to use an appropriate dose of AXs [16].

In order to prevent the bread staling process, the semi-finished bakery products (made after the first baking using the postponed baking method) are refrigerated or frozen [17,18]. The use of cooling or freezing techniques for semi-finished bakery products makes it possible to obtain products with properties similar to those manufactured by traditional methods, such as the preferred volume, crumb moisture and a crispy crust.

Research on the influence of isolated natural and modified arabinoxylans and their structure on the properties of rye bread prepared using the postponed baking method has not been conducted so far. There are indications that the use of AXs in postponed baking may be especially effective [19,20,21,22]. Previous studies using AX-enriched rye flour have shown that rye bread baked with 10% AXs using the postponed baking method had a higher volume and moisture, as well as lower crumb hardness, compared to control bread [4].

The aim of this work was to investigate the influence of rye arabinoxylans of different molar mass (natural and cross-linked) on the properties of rye bread baked using the postponed baking method.

## 2. Materials and Methods

The research materials were rye flours types 720 and 1150 (PZZ Kraków, Poland). Details of the chemical composition of the flours are presented in Table 1.

Arabinoxylan preparations were isolated from rye wholemeal of the Amilo variety (Danko, Poland) produced by a laboratory method (total, water-soluble and water-insoluble AX content of 11.1%, 3.2% and 7.9%, respectively). In this research, laboratory preparations containing unmodified (LP_NM) and cross-linked arabinoxylan (LP_CR) were used. Baker’s acid was purchased from the Bionat company (Krakow, Poland). Yeast (Lesaffre, Marcq-en-Barœul, France) and salt (NaCl, Avantor Performance Materials Poland S.A., Gliwice, Poland) were also used for rye dough preparation.

### 2.1. Isolation and Modification of AXs

#### 2.1.1. Isolation of Rye Water Extractable AXs

Rye wholemeal (100 g) was extracted with 500 mL of 80% *v*/*v* EtOH at 90 °C for 2 h to inactivate cereal enzymes. After cooling, the ethanol solution was decanted, and the sediment was dried at 40 °C overnight, followed by extraction with 2 L of water at 25 °C for 6 h. The suspension was centrifuged, and the supernatant was boiled to coagulate the soluble proteins; then, the sample was cooled and incubated with α-amylase (saliva, Sigma Aldrich, St. Louis, MO, USA) at 37 °C for 2 h. Saliva amylase was applied in order to avoid incomplete inactivation by heating, the residual activity of amylase may cause degradation of starch in the bread crumb. The solution was boiled, and then, for protein removal, 20 g/L celite (diatomaceous earth, Megazyme, Bray, Ireland) was added and the whole suspension was filtered via a Buchner flask. The clear filtrate was poured into a 4-fold solution of ethanol/acetone (1:1). The sediment was centrifuged and then frozen at −18 °C and stored for further modification by cross-linking, or washed twice with ethanol/acetone and twice with acetone. After the last centrifugation, the AX sediment was dried at 50 °C for 2 h. The yield of the unmodified arabinoxylan preparation (LP_NM) was 2.72 g/100 g of rye flour.

#### 2.1.2. Cross-Linking of AXs

Fifteen grams of the frozen sediment was dissolved in 40 mL of deionized water with stirring at 50 °C for 6 h. After this time, the solution was cooled to 25 °C, hydrogen peroxide (1 μg/g of AXs) and peroxidase (Horseradish, Sigma Aldrich) at a concentration of 5 U/g of AXs were added, and the solution was treated for 15 min. The process was stopped by flooding the solution with a 4-fold volume of ethanol/acetone solution (1:1). The AX precipitate was centrifuged and washed twice with ethanol/acetone and twice with acetone to remove all water from the sample. After the last centrifugation, the precipitated cross-linked AXs (denoted as LP_CR) were dried at 50 °C for 2 h. 

### 2.2. Determination of the Monosaccharide Composition of Arabinoxylan Preparations

The sugar profile of the AXs was determined by HPLC after acid hydrolysis of AX preparations according to Buksa et al. [7]. In short, 20 µL of 4 mg/mL hydrolyzed sample solution was injected into an HPLC system (Knauer, Berlin, Germany) equipped with a Sugar SP08010 column (Shodex, Japan). The arabinose-to-xylose ratio (A/X) was calculated by dividing the content of arabinose by the content of xylose. The content of AXs was calculated as the sum of the content of arabinose and xylose and multiplied by a factor of 0.88. The glucan content was calculated by multiplying the glucose content by a factor of 0.9.

### 2.3. Determination of Molecular Properties of Arabinoxylan

The molecular properties of the AXs were determined by SEC according to Buksa et al. [23]. The SEC system (Knauer, Berlin, Germany) was equipped with OHpak SB-806HQ and SB-804HQ (Shodex, Tokyo, Japan) columns and a refractometric detector (Knauer, Germany). A total of 100 mM NaNO_3_ was used as an eluent (to avoid aggregates) at a flow rate 0.6 mL/min. The columns were maintained at 60 °C. The SEC system was calibrated with pullulan standards (Shodex Standard, Macherey-Nagel, Düren, Germany) and arabinose (Sigma-Aldrich). The data were processed by the Eurochrom (Knauer) and Clarity (ver. 4.0.1.700, DataApex) programs.

### 2.4. Determination of Protein Content in Arabinoxylan Preparations

The protein content was determined by Kjeldahl’s method according to AOAC 950.36 (N × 6.25) [24].

### 2.5. Rye Dough Preparation

The doughs were prepared and the rye breads were baked from the tested commercial rye flours (types 720 and 1150) of various ash contents. According to the basic recipe, the rye bread dough was made of 100 g of rye flour (type 720 or 1150), water (30 °C) in an amount determined by a Farinograph at the same dough consistency of 150 BU, 2% non-iodized salt, 3% yeast (*Saccharomyces cerevisiae*) and 8% Bionat baking acid. The above-mentioned recipe was used to prepare the doughs and breads, which included a control sample. The AX preparations were added in an amount of 1% to the flour in powdered form at the beginning of mixing. The dough (both without and with AX addition) was kneaded for a maximum of 12 min in a Farinograph to obtain a consistency of 150 BU.

### 2.6. Dough Analysis

The textural properties of the prepared rye dough were tested using a TAXT Plus (Stable Microsystems, Godalming, UK) textrometer. For this purpose, a dough sample (25 cm^3^) was placed in a cylindrical container with a diameter of 4 cm. The test was performed in at least three replications using a P-20 adapter penetrating the dough at a speed of 1 mm/s to a distance of 1 cm. 

### 2.7. Dough Fermentation and Bread Baking Using the Postponed Baking Method

Rye dough fermentation and bread baking using the postponed baking method were conducted according to Buksa et al. [4]. Pieces (60 g) were formed from the prepared dough and fermented for 70 min at 26–30 °C, after which the breads were baked by putting the dough pieces in an oven chamber heated to 160 °C, gradually increasing the temperature to 190 °C and keeping them at 190 °C for 3 min, without allowing the crust to color. The baked breads were then cooled to room temperature and frozen in a shock freezer at −60 °C until the temperature reached −18 °C inside the loaves. The frozen breads were stored in the freezer (−18 °C) for 2 weeks. The 14-day storage period for semi-finished bakery products was also used in other research [4,25,26], and Bárcenas et al. [27] showed that beyond 14 days of storage, there is an asymptotic trend in bread crumb hardness, and this parameter is responsible for staling. After thawing for 60 min, the breads were baked again at 230 °C for 17 min in a Miwe Condo modular electric oven, type C-52.

### 2.8. Determination of the Properties of Breads Baked by the Postponed Method without and with a Share of AX Preparations

Baking loss was determined according to [25]. 

Bread volume was measured by a 3-dimensional laser-based Volscan Profiler scanner (Stable Microsystems, Godalming, UK), according to the manufacturer’s manual. The specific bread volume was calculated by converting the result of the bread volume measurement into bread volume obtained from 100 g of flour. 

Texture profile analysis was conducted using a TAXT Plus (Stable Microsystems, Godalming, UK) textrometer with software Texture Exponent (ver. 3.0.5.0, Stable Microsystems, England) and a TPA test (examining slices of bread 3 cm thick obtained from three breads) [28]. A sample of bread crumb, taken from the center of the loaf with a height of 3 cm, was pressed by a P-20 aluminum compression platen with a diameter of 15 mm at a compression rate of 5 mm/s to a distance of 1 cm in two cycles with a 5 s delay. 

The bread crumb moisture was determined by a gravimetric method by AOAC 925.10 [24]. A crumb sample of approx. 1 g was placed in a dish, after weighing both the sample and the dish (separately). The sample was dried at 130 °C for 1 h, and the dish was weighed with the sample. The percentage of crumb moisture was determined from the difference in sample weight before and after drying.

### 2.9. Statistical Analysis

All analyses were performed at least in triplicate. The data were statistically evaluated using an analysis of variance (ANOVA). Statistica v. 9.0 software (StatSoft, Inc., Tulsa, OK, USA) was used for the statistical analysis.

## 3. Results and Discussion

The basic chemical composition of the unmodified (LP_NM) and cross-linked (LP_CR) water-soluble arabinoxylan preparations is shown in Table 2.

The composition of sugar residues released after acid hydrolysis of the AX preparations was similar. The preparations contained mainly xylose and arabinose originating from the AXs (estimated AX content in preparations: 82–84%; A/X ratio: 0.73), which was consistent with data from other studies [7,11,29]. The protein content in the AX preparations was similar to the data presented in the literature, where the content of this ingredient ranged from 3.8% to 22.3% [7,30,31,32]. Figure 1 shows the molar mass distributions of unmodified and cross-linked AXs. In the case of cross-linked AXs (LP_CR), a significant shift in the distribution towards higher molar masses was observed compared to unmodified arabinoxylans (LP_NM), confirming the success of the modification (cross-linking). Moreover, the modification was very effective, as evidenced by the clear shift towards higher molar masses. Based on the molar mass distribution (Figure 1), the following molecular parameters of the arabinoxylans were determined: weight average molar mass (Mw), number average molar mass (Mn) and dispersity (Ð). The molar mass of the unmodified AXs (LP_NM) was similar to that reported in the literature, where this parameter was in the range of 40,000–500,000 g/mol [7,11,15,33]. The molar mass of cross-linked AXs (LP_CR) was much higher compared to the data in previous literature reports, according to which it was 505,000 g/mol [23]. Such a high molar mass of the LP_CR preparation indicates that the modification (cross-linking) process of AXs was particularly effective. The rye grain used for the extraction of arabinoxylans likely contained AXs with a high amount of a side substituent, such as ferulic acid, which is involved in cross-linking the AX molecules [30,34].

To examine the influence of AXs on the properties of rye dough and bread baked using the postponed baking method, AX preparations were applied at a 1% share (1% of AX preparation was used in place of flour). According to the data presented in the literature, the 0.5% dose was proven to be too small [7]. On the other hand, preliminary research indicated that a higher share of AXs was excessive, leading to over-adhesiveness of the dough and difficulties in its processing (dough handling). Moreover, a high share of AXs is not economically justified due to the high cost of AX preparations, especially cross-linked ones. The establishment of a method for adding AX preparations to flour for dough was preceded by preliminary research on their solubility, in order to choose the optimal method before use. This research showed that the best way to dose these preparations is to add them to flour in powdered form, at the beginning of mixing.

Research on the influence of AXs on the properties of rye bread was carried out using the two most popular rye flours with lower (T720) and higher (T1150) ash content. These flours differed in their chemical composition (Table 1). T1150 flour contained more ash, fiber, AXs and protein, but less starch, compared to T720 flour.

The influence of a 1% share of AX preparation on the addition of water to the dough and its textural properties is shown in Table 3. 

A higher addition of water to the dough required to achieve an optimal consistency of 150 BU was observed using rye flour type 1150 compared to rye flour type 720 (Table 3). The ash content of each type of flour affects its water absorption. However, the more ash in the flour, the higher its hydration capacity [35,36].

The share of both the tested AX preparations increased the amount of water needed to achieve a dough with a final consistency of 150 BU for both rye flour types 720 and 1150. The addition of water was proportional to the molar mass of AXs. A 1% share of AXs (LP_CR) in rye flour type 720 resulted in the highest water addition to the dough needed to achieve an optimal consistency of 150 BU, as the amount of water added to the flour increased by 18%, compared to the control sample. Similarly, although to a smaller extent than in the case of flour type 720, the share of cross-linked AXs (LP_CR) also resulted in a higher addition of water to flour type 1150. The observed increase in water addition to the dough, particularly with the use of cross-linked AXs, was consistent with data from the literature on a model rye dough with a share of AXs [7]. In addition, the observed stronger effect of the share of both AX preparations tested, especially the cross-linked one, on the addition of water to doughs made from flour 720 was due to the smaller content of fiber, so this also applied to arabinoxylan in flour type 720, compared to flour type 1150 (Table 1). 

A direct consequence of the increased addition of water to flour is the dough yield, which was calculated and is presented in Table 3. The type of flour used for bread baking has a significant (*p* < 0.05) effect on the dough yield, which was confirmed by the results [35,36]. The use of type 1150 flour resulted in higher dough yields compared to dough made from flour of type 720. The influence of AX share on rye dough yield was a consequence of the increased water addition and followed the same trend. The highest dough yield from both types of flours was recorded using a share of the preparation of cross-linked AXs (LP_CR).

Postponed bread baking is usually carried out in the bakery industry using appropriate machinery, so the textural properties of the dough used in this technology are particularly important [37]. Table 3 presents the results of the hardness of doughs without and with AX addition determined by a backward extrusion test. The higher the dough hardness, the more difficult it is to knead, which poses a problem for technologists [38,39,40].

Comparing the control samples, the type of rye flour did not significantly affect the dough hardness parameter, which was consistent with data presented in the previous literature [41].

It was observed that with a dough consistency of 150 BU, a 1% share of LP_NM tended to result in a decrease in the hardness of the dough made from rye flour types 720 and 1150 compared to the appropriate control sample. A 1% share of the LP_CR preparation in the type 720 rye flour dough did not significantly (*p* < 0.05) affect this parameter, but an increase in the hardness of the dough with the LP_CR preparation was noted. The tendency to reduce dough hardness with the use of unmodified AXs, at the same dough consistency of 150 BU, was confirmed by data from the literature that describe a similar effect of AXs [42]. 

Determining the adhesiveness of the dough helps to assess the force required to overcome its viscosity [38,39,40]. Too high a dough’s adhesiveness during processing is not preferable, as it causes the dough to stick excessively to machine surfaces [37].

When comparing control samples (without the addition of AXs), it was observed that the adhesiveness of dough made from rye flour type 720 did not differ significantly (*p* < 0.05) from that of dough made from rye flour type 1150 (Table 3), which was consistent with data presented in the previous literature [41].

As a result of the application of both preparations of AXs, a tendency to reduce the adhesiveness of the dough of rye flour type 720 was observed. Only LP_CR preparation added to rye flour type 1150 showed a tendency to increase this parameter, compared to dough without added AXs. The influence of unmodified AXs (LP_NM) on the reduction in the adhesiveness was also observed for rye dough prepared in traditional baking [42].

In the next step, the postponed baking of breads was carried out using commercial rye flours of types 720 and 1150 without and with 1% (used in place of part of the flour) AX preparations isolated by laboratory method from rye flour: unmodified (LP_NM) and cross-linked (LP_CR). The results of determining the properties of breads obtained in postponed baking are shown in Table 4.

The baking loss of rye bread baked using the postponed baking method with type 720 flour was about 1.3% lower compared to rye bread made with type 1150 flour, which was compatible with data in the existing literature [4]. Compared to the dough made from type 1150 flour, dough made from type 720 flour (low-fiber, Table 1) required lower water addition (Table 3) to achieve the optimal 150 BU consistency. Additionally, less water evaporated from this dough during baking resulting in a lower baking loss of the bread [43]. 

The influence of the share of AX preparations on the baking loss of rye bread made from type 720 flour was inconclusive (Table 4). The 1% share of the LP_NM preparation had no significant effect, while the share of LP_CR resulted in an increase in baking loss for bread made from rye flour type 720 baked by the postponed baking method compared to bread without AX addition. The share of the LP_NM preparation resulted in a decrease in the baking loss of bread made from rye flour type 1150 baked by the postponed baking method, while the LP_CR preparation led to an increase, compared to bread without the addition of AXs. The observed higher baking losses of breads with LP_CR for both types of rye flours can be explained by the greater addition of water to the doughs. Breads with LP_CR that contained more water, which evaporated to a greater extent during baking, resulted in higher baking losses.

Rye bread made from type 720 flour baked using the postponed baking method was characterized by approximately a 24% higher volume (BV, Table 4) compared to rye bread made from type 1150 flour baked using the same method. The observed dependence of bread volume on flour type was typical [4,42,44,45]. 

The share of both the preparation of unmodified (LP_NM) and cross-linked AXs (LP_CR) had a positive effect on the volume of rye bread (Table 4) baked from both types of flours using the postponed baking method compared to bread without added AXs. The highest increase in the volume (by about 20%) of rye breads baked by the postponed baking method was observed as a result of the addition of cross-linked AXs (LP_CR) to both types of rye flours compared to breads without added AXs. A similar effect of AXs has been reported for breads baked by the traditional method [41]. The higher volume of breads with AXs has been explained by the content and effect of water-soluble AXs both already present in the flour and added. Arabinoxylans strengthen the gas bubbles in the dough and thus allow them to grow more, which results in a higher volume of baked bread [41,42,45].

Specific bread volume (SBV), defined as the total volume of bread obtained from 100 g of flour, is an important parameter for the baking industry. It considers not only the volume of a loaf from a certain amount of dough, but also the flour’s water absorption and, consequently, the dough yield. Analogous to bread volume, SBV was significantly (*p* < 0.05) higher for rye bread made from 720 type flour, compared to rye bread made from 1150 type flour (Table 4). 

The share of both AX preparations increased (*p* < 0.05) the specific volume of rye breads baked by the postponed baking method from both types of flour (Table 4). The 1% share of the LP_CR preparation had a higher influence on increasing the SBV compared to the LP_NM preparation. The highest increase (by about 23%) in SBV was observed as a result of using LP_CR for postponed baked rye bread made from 1150 flour compared to bread without AX addition.

Rye bread made from flour with a higher ash content (type 1150) had a higher (*p* < 0.05) crumb moisture content compared to rye bread made from type 720 flour. This result was correlated with a higher addition of water to the flour at the dough preparation stage. Flour of the 1150 type is characterized by higher water absorption, as there are naturally more arabinoxylans in 1150 type flour than in 720 type flour [45].

The 1% share of both tested AX preparations positively increased the crumb moisture of rye breads baked by the postponed baking method from both type 720 and 1150 flour (Table 4). The crumb moisture content of rye breads baked by the postponed baking method from type 720 flour with LP_NM and LP_CR preparations was higher by approximately 1.7% and 3.8%, respectively, compared to bread without AXs (control sample). The use of LP_NM and LP_CR preparations for breads made from 1150 flour led to higher crumb moisture content of the breads, by 1.3% and 2%, respectively, compared to breads without added AXs (Table 4). In the crumb of breads without added AX preparations, water was bound only by the flour components, mainly by gelatinized starch. While in the crumb of breads with an AX share, water was additionally bound by added AXs [4].

The highest increase in crumb moisture was observed when a 1% share of cross-linked AX preparation (LP_CR) was used to bake postponed rye bread from type 720 flour. The particularly beneficial effect of cross-linked AXs on bread crumb moisture content has also been indicated by data from the literature [4,22,42]. Cross-linked AXs with a high molar mass used in the baking of rye bread using the postponed baking method caused a high amount of water to be bound during dough formation, which affected the final moisture content of the bread crumb [4]. Moreover, the addition of other hydrocolloids to wheat bread baked using the postponed baking method increased this parameter [22,46,47,48].

The texture parameters of the bread crumb are shown in Table 4. From the parameters of the TPA analysis, only hardness, adhesiveness, springiness and cohesiveness were found to be the most important features. 

The crumb hardness of bread made from type 1150 flour was higher compared to bread made from 720 flour, which was compatible with data presented in the literature [4]. In the baking of traditional rye bread from flours of types 1150 and 720, higher crumb hardness was observed in bread baked from flour type 1150, which was related to the higher content of bran particles [41].

As a result of the share of both AX preparations tested, a reduction in crumb hardness was observed in breads baked from both 720 and 1150 flour using the postponed baking method compared to breads without AXs (Table 4). The highest reduction in crumb hardness was observed when the LP_NM preparation was applied to rye bread baked by the postponed baking method from type 1150 flour, as its 1% share reduced the hardness of the bread crumb by approx. 32% compared to bread without AX addition. The positive effect of AXs, as well as other hydrocolloids, on reducing the hardness of the crumb of rye bread, and wheat bread baked by the postponed baking method, has been reported in the literature [4,22,49].

Rye bread from type 720 flour baked by the postponed baking method was characterized by a lower crumb adhesiveness of more than 3.5 times compared to bread from type 1150 flour. A similar correlation has also been indicated by data from the literature on rye breads baked by the postponed baking method [4], as well as by the traditional method [41]. It can be speculated that the higher content of insoluble AXs in flour of a higher type affects the adhesiveness of the crumb of rye bread baked by the postponed baking method. 

The 1% share of both preparations did not significantly affect the crumb adhesiveness of rye breads made from both types of flour compared to the corresponding breads without AX addition (Table 4). Previous studies have reported that the high crumb adhesiveness of rye bread is correlated with the content of natural as well as added arabinoxylans [50]. The solubility of the AXs present in the flour is probably also important.

Rye bread from type 720 flour baked by the postponed baking method was characterized by 15.5% higher crumb springiness compared to bread from type 1150 flour, which could be due to the different chemical composition of the flours. AX preparations did not significantly affect the crumb springiness of rye breads baked by the postponed baking method from both types of flour compared to breads without AX addition. 

Rye bread made from type 720 flour baked using the postponed baking method had 10.5% higher crumb cohesiveness compared to bread made from type 1150 flour, which was a consequence of the different chemical composition of the flours. 

A 1% share of AX preparations increased the crumb cohesiveness (Table 4) of rye breads baked by the postponed baking method from type 720 flour compared to bread without AX addition. In the case of bread made from type 1150 rye flour with a 1% share of LP_NM preparation, the cohesiveness of the crumb did not change significantly (*p* < 0.05) compared to bread without the addition of AXs, and the preparation of cross-linked AXs (LP_CR) increased the cohesiveness of the crumb of rye bread made from type 1150 flour.

This research conducted at the laboratory scale shows the need for further research on the efficient isolation of AXs, the modification of AXs’ structure, and their application in rye bread baking using the postponed baking method. 

## 4. Conclusions

The 1% share of AX preparations increased the addition of water to rye flour. A preparation of cross-linked AXs with a very high molar mass of 1,158,980 g/mol was particularly effective in increasing water addition and therefore dough yield.

Regardless of the type of flour, the 1% share of both AX preparations had a positive effect on increasing the volume of rye breads baked using the postponed baking method compared to breads without the addition of AXs. The preparation of cross-linked AXs (LP_CR) proved to be particularly effective in increasing the volume of bread baked using the postponed baking method.

Regardless of the type of flour, a 1% share of both AX preparations had a positive effect on increasing the crumb moisture of rye bread baked using the postponed baking method compared to bread without the addition of AXs. The preparation of cross-linked arabinoxylans (LP_CR) had the most effective impact on increasing the moisture of the rye bread crumb.

The 1% share of both AX preparations had a positive effect on reducing the hardness of the crumb of rye breads baked by the postponed baking method compared to breads without AX addition. 

The results of this study provide a basis for the development of innovative rye breads with a share of AXs of an appropriate molar mass. The major limitation of this research was the lack of commercially available AX preparation. Commercially available AX preparation could be used on a bigger scale and the research could be extended. 

## Figures and Tables

**Figure 1 foods-13-02482-f001:**
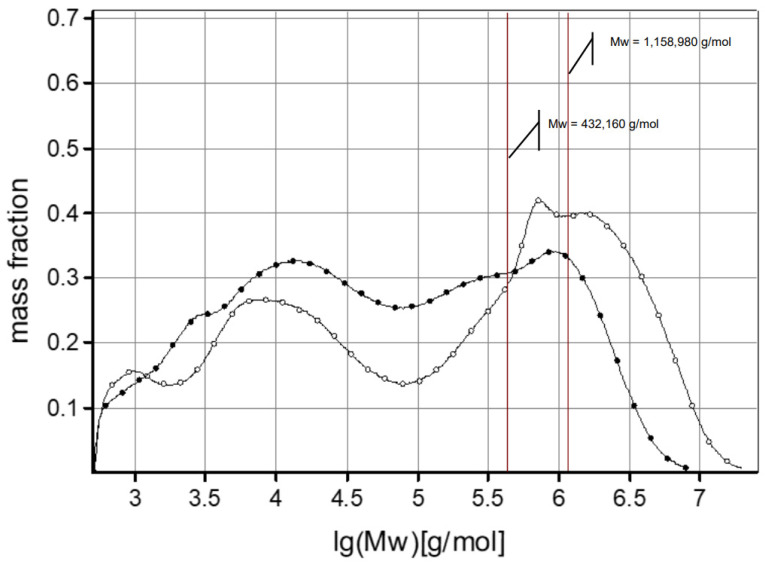
Molecular mass distribution profiles of native LP_NM (-●-) and cross-linked LP_CR (-○-) water-soluble arabinoxylan.

**Table 1 foods-13-02482-t001:** Chemical composition of rye flours.

Proximate Composition	Rye Flour Type 720	Rye Flour Type 1150
Moisture [%]	12.6 ± 0.4 ^a^	12.4 ± 0.6 ^a^
Starch [%]	79.5 ± 0.9 ^b^	64.4 ± 1.4 ^a^
Protein [%]	6.0 ± 0.5 ^a^	10.6 ± 0.7 ^b^
Total dietary fiber [%]	6.9 ± 0.5 ^a^	12.8 ± 0.7 ^b^
Ash [%]	0.724 ± 0.027 ^a^	1.028 ± 0.019 ^b^
Arabinoxylans [%]	4.2 ± 0.4 ^a^	5.7 ± 0.5 ^b^

Mean values marked with the same letters in particular rows are not statistically significantly different at *p* < 0.05.

**Table 2 foods-13-02482-t002:** Basic chemical composition and molecular properties of water-soluble, unmodified (LP_NM) and cross-linked (LP_CR) preparations of arabinoxylan.

	LP_NM	LP_CR
Arabinose [%]	39.7 ± 0.4 ^a^	40.1 ± 0.7 ^a^
Xylose [%]	54.1 ± 0.9 ^a^	55.2 ± 0.8 ^a^
Glucose [%]	4.3 ± 0.4 ^a^	4.1 ± 0.7 ^a^
Mannose [%]	3.3 ± 0.4 ^a^	3.3 ± 0.4 ^a^
A/X ratio HPLC	0.73 ^a^	0.73 ^a^
Arabinoxylan content [%]	82.5 ± 1.2 ^a^	83.9 ± 1.4 ^a^
Glucan content [%]	6.8 ± 0.7 ^a^	6.7 ± 1.0 ^a^
Protein content (N × 6.25) [%]	9.7 ± 0.5 ^a^	9.5 ± 0.6 ^a^
Molar mass of arabinoxylan		
M_w_ * [g/mol]	432,160 ^a^	1,158,980 ^b^
M_n_ * [g/mol]	7 850 ^a^	8 140 ^b^
Ð*	55.1 ^a^	142.4 ^b^

* M_w_/M_n_—weight/number average molar mass; Ð—dispersity. Mean values marked with the same letters in particular rows are not statistically significantly different at *p* < 0.05.

**Table 3 foods-13-02482-t003:** The influence of AX preparations on the addition of water to the dough, the yield of the dough and its textural properties.

Parameters	Flour Type 720			Flour Type 1150		
	CRF *	CRF + LP_NM **	CRF + LP_CR ***	CRF *	CRF + LP_NM **	CRF + LP_CR ***
Water addition [mL]	39.3 ± 0.1 ^a^	42.4 ± 0.4 ^b^	46.5 ± 0.2 ^c^	52.5 ± 0.4 ^d^	53.8 ± 0.1 ^e^	55.3 ± 0.1 ^f^
Yield of dough [%]	139.3± 0.1 ^a^	142.4 ± 0.4 ^b^	146.5 ± 0.2 ^c^	152.5 ± 0.4 ^d^	153.8 ± 0.1 ^e^	155.3 ± 0.1 ^f^
Hardness of dough [N]	4.2 ± 0.4 ^ab^	3.6 ± 0.2 ^a^	4.4 ± 0.0 ^b^	4.2 ± 0.3 ^ab^	4.0 ± 0.1 ^ab^	5.5 ± 0.2 ^c^
Adhesiveness of dough [N·s]	−611.4 ± 61.2 ^a^	−525.9 ± 30.9 ^a^	−565.3 ± 28.1 ^a^	−650.7± 72.5 ^ab^	−602.5 ± 33.7 ^a^	−782.1 ± 84.6 ^b^

* CRF—commercial rye flour, control bread; ** commercial rye flour with 1% share of unmodified preparation of arabinoxylan; *** commercial rye flour with 1% share of cross-linking preparation of arabinoxylan. Mean values marked with the same letters in particular rows are not statistically significantly different at *p* < 0.05.

**Table 4 foods-13-02482-t004:** Properties of breads baked by the postponed baking method.

Parameters	Flour Type 720			Flour Type 1150		
	CRF *	CRF + LP_NM **	CRF + LP_CR ***	CRF *	CRF + LP_NM **	CRF + LP_CR ***
Baking loss [%]	13.1 ± 0.1 ^b^	13.0 ± 0.3 ^b^	17.7 ± 0.2 ^e^	14.4 ± 0.0 ^c^	12.3 ± 0.1 ^a^	15.3 ± 0.2 ^d^
Bread volume [cm^3^]	90.5 ± 1.0 ^c^	94.5 ± 1.0 ^d^	105.0 ± 0.0 ^e^	73.0 ± 1.2 ^a^	79.0 ± 1.2 ^b^	88.0 ± 1.2 ^c^
Specific bread volume [cm^3^]	210.1 ± 2.3 ^c^	224.3 ± 2.4 ^d^	256.4 ± 0.0 ^e^	185.5 ± 3.1 ^a^	202.5 ± 3.1 ^b^	227.8 ± 3.1 ^d^
Crumb moisture [%]	49.9 ± 0.3 ^a^	51.6 ± 0.2 ^b^	53.7 ± 0.1 ^c^	55.1 ± 0.4 ^d^	56.4 ± 0.1 ^e^	57.1 ± 0.2 ^f^
Hardness [N]	8.2 ± 0.0 ^ab^	7.3 ± 0.7 ^a^	7.6 ± 0.2 ^a^	12.1 ± 1.0 ^c^	8.2 ± 0.5 ^ab^	9.5 ± 0.2 ^b^
Adhesiveness [N·s]	−15.5 ± 2.4 ^a^	−13.5 ± 4.5 ^a^	−6.1 ± 0.9 ^a^	−54.7 ± 7.0 ^bc^	−40.1 ± 8.7 ^b^	−61.0 ± 5.2 ^c^
Springiness [−]	0.974 ± 0.058 ^b^	0.914 ± 0.035 ^ab^	0.967 ± 0.012 ^b^	0.823 ± 0.051 ^a^	0.861 ± 0.019 ^ab^	0.875 ± 0.057 ^ab^
Cohesiveness [−]	0.693 ± 0.001 ^b^	0.741 ± 0.005 ^c^	0.805 ± 0.013 ^d^	0.620 ± 0.019 ^a^	0.606 ± 0.007 ^a^	0.634 ± 0.007 ^b^

* CRF—commercial rye flour, control bread; ** commercial rye flour with 1% share of unmodified preparation of arabinoxylan; *** commercial rye flour with 1% share of cross-linking preparation of arabinoxylan. Mean values marked with the same letters in particular rows are not statistically significantly different at *p* < 0.05.

## Data Availability

The original contributions presented in the study are included in the article, further inquiries can be directed to the corresponding author.

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
