# Peer review of "The Influence of Arabinoxylan of Different Molar Masses on the Properties of Rye Bread Baked by the Postponed Baking Method"

_foods, 2024, doi:10.3390/foods13162482_

Round 1

Reviewer 1 Report

Comments and Suggestions for Authors

This study gives the evidence that the natural water-soluble and cross-linked arabinoxylan preparations both had positive effect on reducing the hardness of the crumb of rye breads baked by the postponed baking method. This idea is new and the experimental design is very interesting. Suggestions,

1)     Line 73-117, the producing company of enzymes (alpha-amylase, and peroxidase) used in this study should be given. Line 78-79, why was a glucoamylase not used? What was the effect of cellite?

2)     Line 97, what was the chromatography column used for the separation of arabinose, xylose, mannose, and glucose?

3)     Line 112, the pH of the eluent had better be given? Why was sodium nitrate solution used as the eluent?

4)     In figure 1, the molecular weight marker should be marked.

5)     In table 2, 0% and 1% should be noted. The unit of the dough adhesiveness should be given.

6)     Line 155, the method of moisture content in bread crumb had better be described briefly.

7)     Why was the chemical composition of commercial rye flours type 720 and 1150 shown in a supplementary table? It is better to insert in the text. The units of items in this table should be given.

8)     How to use water soluble arabinoxylan and cross-linked arabinoxylan to prevent the staling process in breads and noodles is paying close attention in research and in practice. This study gave a study method. In the abstract and conclusion sections, the further research direction should be given.

Author Response

This study gives the evidence that the natural water-soluble and cross-linked arabinoxylan preparations both had positive effect on reducing the hardness of the crumb of rye breads baked by the postponed baking method. This idea is new and the experimental design is very interesting. Suggestions,

1)     Line 73-117, the producing company of enzymes (alpha-amylase, and peroxidase) used in this study should be given. Line 78-79, why was a glucoamylase not used? What was the effect of cellite?

Answer: Information about the producing company - Sigma Aldrich - was added. Saliva amylase was applied to avoid incomplete enzyme inactivation by heating. The residual activity of amylase may cause the degradation of starch in the bread crumb. We have tested different enzymes and under the conditions applied saliva amylase was the most effective and easy to inactivate.

Celite (diatomaceous earth, Megazyme) was used for protein removal (by complexation).

Appropriate changes were included in the text.

2)     Line 97, what was the chromatography column used for the separation of arabinose, xylose, mannose, and glucose?

Answer: The separation was done using the Sugar SP08010 column (Shodex, Japan). Appropriate changes were included.

3)     Line 112, the pH of the eluent had better be given? Why was sodium nitrate solution used as the eluent?

Answer: Using 0.1 M NaNO3 solution as an eluent for SEC separations is rather common and recommended by the column manufacturer (Shodex, Japan). Providing pH is rather unusual, this will be neutral. Salt solution as an eluent is necessary to avoid aggregation of big molecules that may have a strong impact on the determined molar masses of hydrocolloids. A short explanation was added to the manuscript.

4)     In figure 1, the molecular weight marker should be marked.

Answer: Appropriate changes were included.

5)     In table 2, 0% and 1% should be noted. The unit of the dough adhesiveness should be given.

Answer: Tables were reorganized. Appropriate changes were included.

6)     Line 155, the method of moisture content in bread crumb had better be described briefly.

Answer: Appropriate changes were included.

7)     Why was the chemical composition of commercial rye flours type 720 and 1150 shown in a supplementary table? It is better to insert in the text. The units of items in this table should be given.

Answer: Appropriate changes were included.

8)     How to use water soluble arabinoxylan and cross-linked arabinoxylan to prevent the staling process in breads and noodles is paying close attention in research and in practice. This study gave a study method. In the abstract and conclusion sections, the further research direction should be given.

Answer: Thank you for this comment. Due to the word count limit, it was not possible to add such information to the abstract. However, we have added appropriate information at the end of the discussion and in the conclusions.

Reviewer 2 Report

Comments and Suggestions for Authors After reading the manuscript "The influence of arabinoxylan of different molar mass on the 2 properties of rye bread baked by the postponed baking method", I realized that the manuscript showed in some parts the scientific rigour wanted, but in other parts I have missed it.

The authors have presented critical evaluation only in some paragraphs.

The references are not exactly current, besides Introduction, rationale, Methods and Results should be reviewed.

Thats why I have written some suggestions below in an attempt to improve the paper.

L.39- "the improvement of bread quality" - Please, which improvements would be?

L.50- "these techniques" - Which techniques ? Please, be clearer.

L.49 and L.54- You use 2 nomenclatures: "semi-finished bakery" and "postponed baking method" - are they used synonymously? You need to clarify. 

L.60-  "rye" arabinoxylans - Do you really need "rye" twice ?

L.121- It would be interesting to show images of the loaves so that we can see the differences.

Did you carry out a sensory analysis? I think that if you have this result, it would be important to include it in the evaluation.

L. 122- What type of bread ? We have so many options.

L.124- Which flour ? refined salt ? Which yeast ? pls, be more clearer, more details are necessary.

L.128- What about treatment 0%?

L.142- Which author? Does this method always have to be for 2 weeks? This was not mentioned in the Introduction, but it seems very relevant to me.

L.145- I evaluate a lot of papers and I don't remember seeing this format in topics in the material and methods.

L.153- You mention below that the replications were mostly in triplicate, but what about the texture?

L.155- Crumb moisture. pls, use the same names in the whole paper.

L.187- I missed the statistical analysis of this results in this table.

L.210- I don't believe the way you've presented the tables is very useful, perhaps moving the 0% and 1% to the left column would make them easier to understand.

L.294 Table 3- 224.3± 2.4a    210.1± 2.3b   What about letter C ? I do not understand. Please, check all the results.

L.294- Table 3 and 4 could be combined into one. I don't understand why this presentation with blank spaces makes it difficult to understand visually at first. For example CRF + LP_NM** Hardness [N] and so on ?

L.398- Don't end the paper with a table before the conclusion

L.402- What are the limitations of this study ?

L.403- I haven't seen the conclusion in topics as the authors have presented it, usually the authors write it in paragraphs.

Comments on the Quality of English Language

Minor editing of English language required

Author Response

After reading the manuscript "The influence of arabinoxylan of different molar mass on the 2 properties of rye bread baked by the postponed baking method", I realized that the manuscript showed in some parts the scientific rigour wanted, but in other parts I have missed it.

The authors have presented critical evaluation only in some paragraphs.

The references are not exactly current, besides Introduction, rationale, Methods and Results should be reviewed.

Answer: Appropriate changes were included.

Added publications:

Borczak, B.; Sikora, E.; Sikora, M.; Kapusta-Duch, J.; Hrusavov, D. Effect of bake-off technology and addend sourdough on in vitro glycemic index and on content of Total starch and polyphenols in wheat flour rolls. Å»NTJ. 2014, 5(96), 155–167 (in Polish).

Bárcenas, M. E.; Haros, M.; Benedito, C.; Rosell, C. M. Effect of freezing and frozen storage on the staling of part-baked bread. Food Res. Int. 2003, 36(8), 863–869. doi:10.1016/s0963-9969(03)00093-0. 

Replaced publications:

  • line 164

Ambroziak Z. Piekarstwo i Ciastkarstwo. WNT 1988, Poland.

replaced to:

Murat KaraoÄŸlu, M.; Gürbüz Kotancilar, H. Effect of partial baking, storage and rebaking process on the quality of white pan bread. JFST. 2006, 41(s2), 108–114. doi:10.1111/j.1365-2621.2006.01432.x.

  • line 187

Hartmann, G.; Piber, M.; Koehler, P. Isolation and chemical characterization of water-extractable arabinoxylans from wheat and rye during breadmaking. Eur. Food Res. Technol. 2005, 221, 487–492.

replaced to:

Rosicka-Kaczmarek, J.; Komisarczyk, A.; Nebesny, E. Heteropolysaccharide preparations from rye and wheat bran as sources of antioxidants. J. Cereal Sci. 2018, 81, 37–43. doi:10.1016/j.jcs.2018.03.013.

  • line 292, 308

Korzeniowska-Ginter, Renata. Wplyw jakosci maki tortowej na cechy biszkoptu. Å»ywność Nauka Technologia Jakość. Suplement. 2006, 1, 35-43.

replaced to:

Szwedziak, K.; PolaÅ„czyk, E.; DÄ…browska-Molenda, M.; Nowaczyk, M. Analysis of quality of selected wheat flour types®. Technological Progress in Food Processing. 2018, 2, 5-8. (in Polish).

  • line 627

Buksa, K. Application of model bread baking in the examination of arabinoxylan—protein complexes in rye bread. Carbohydr. Polym. 2016148, 281–289.

replaced to:

Arufe, S.; Chiron, H.; Doré, J.; Savary-Auzeloux, I.; Saulnier, L.; Della Valle, G. Processing & rheological properties of wheat flour dough and bread containing high levels of soluble dietary fibres blends. Food Res. Int. 2017, 97, 123–132. doi:10.1016/j.foodres.2017.03.040. 

Thats why I have written some suggestions below in an attempt to improve the paper.

L.39- "the improvement of bread quality" - Please, which improvements would be?

Answer: The improvement of bread quality is considered as high bread volume, low hardness, and high moisture of the bread crumb. Appropriate changes were included.

L.50- "these techniques" - Which techniques ? Please, be clearer.

Answer: Appropriate changes were included.

L.49 and L.54- You use 2 nomenclatures: "semi-finished bakery" and "postponed baking method" - are they used synonymously? You need to clarify.

Answer: By semi-finished bakery products we mean products made after the first baking (partly baked dough) using the postponed baking method. An appropriate explanation was added.

L.60-  "rye" arabinoxylans - Do you really need "rye" twice ?

AB: Appropriate changes were included.

L.121- It would be interesting to show images of the loaves so that we can see the differences.

Answer: We decided not to add the images of the loaves because of the rather poor quality of the images. The images wouldn't bring anything new to the discussion. Bread volume was measured by the 3-dimensional laser-based scanner Volscan Profiler (Stable Microsystems, Godalming, UK), more accurately than with seed displacement method.

Did you carry out a sensory analysis? I think that if you have this result, it would be important to include it in the evaluation.

Answer: The AX preparations were produced at a laboratory scale (an inefficient and time-consuming process) therefore we didn’t have sufficient amount to perform baking at a larger scale which is necessary to carry out a sensory analysis. 

  1. 122- What type of bread ? We have so many options.

Answer: We meant rye bread. Appropriate changes were included.

L.124- Which flour ? refined salt ? Which yeast ? pls, be more clearer, more details are necessary.

Answer: Appropriate changes were included.

L.128- What about treatment 0%?

Answer: The paragraph was reorganized and appropriate changes were included.

L.142- Which author? Does this method always have to be for 2 weeks? This was not mentioned in the Introduction, but it seems very relevant to me.

Answer: The 14-day storage period for bakery semi-finished products was also used in other research [Borczak et al., 2014, Murat KaraoÄŸlu&Gürbüz 2006] and Bárcenas et al., 2003 showed that beyond 14 days of storage, there is an asymptotic trend in bread crumb hardness, and this parameter is responsible for staling. Appropriate changes and explanations were included.

L.145- I evaluate a lot of papers and I don't remember seeing this format in topics in the material and methods.

Answer: Appropriate changes were included.

L.153- You mention below that the replications were mostly in triplicate, but what about the texture?

Answer: For all variants tested the slices were obtained from three breads. This way three independent samples were analyzed. Appropriate changes were included.

L.155- Crumb moisture. pls, use the same names in the whole paper.

Answer: Appropriate changes were included.

L.187- I missed the statistical analysis of this results in this table.

Answer: Appropriate changes were included.

L.210- I don't believe the way you've presented the tables is very useful, perhaps moving the 0% and 1% to the left column would make them easier to understand.

Answer: Appropriate changes were included.

L.294 Table 3- 224.3± 2.4a    210.1± 2.3b   What about letter C ? I do not understand. Please, check all the results.

Answer: Thank you for pointing this out. The statistic was checked and corrected. The discussion was also updated.

L.294- Table 3 and 4 could be combined into one. I don't understand why this presentation with blank spaces makes it difficult to understand visually at first. For example CRF + LP_NM** Hardness [N] and so on ?

Answer: Thank you. Appropriate changes were included.

L.398- Don't end the paper with a table before the conclusion

Answer: Appropriate changes were included.

L.402- What are the limitations of this study ?

Answer: Thank you for this comment. The major limitation of this research is the lack of commercially available AX preparation. Commercially available AX preparation could be used on a bigger scale and the research could be extended to for example other postponed baking methods or sensory analysis. We have added an appropriate information at the end of the discussion and in the conclusions.

L.403- I haven't seen the conclusion in topics as the authors have presented it, usually the authors write it in paragraphs.

Answer: Appropriate changes were included.

Comments on the Quality of English Language

Minor editing of English language required

Answer: English language has been checked and revived.

Round 2

Reviewer 2 Report

Comments and Suggestions for Authors Dear authors,    After another evaluation of the manuscript, I realized some improvement in the quality of the paper. The authors have accepted some of my requests.  English is always useful to ask a native speaker for a final appreciation. They added more authors to better substantiate the paper  and improved tables.  But I still have some comments as you can see below:  

L. 68 - Table 1- I suggest changing the letters that differentiate the averages in superscript. Besides, the cell with the proximate composition is in blank and should have a title.

L.124- which temperature of the water ? In my opinion it is an important information 

L.132- Please, check the writting of texturometer. I missed the details about the probe, speed, replicants ... 

L.136- I think it is a pity that you decided  to not include picture of the breads, usually pictures bring more understanding and enchantment.

L.139- This methodology should have author/year.

L.146- This methodology  also should have author/year.

L.155- Please, check the writting of texturometer. I missed the details about the probe, speed,  ... 

L.156- "from three breads " -  Do you think it is enough for a texturometer ?

L.215 -Table 3 -  I suggest changing the letters that differentiate the averages in superscript.

How come ? Hardness of the dough [N]    4.2 ± 0.4abc  4.2± 0.3bc

Adhesiveness of the dough [N·s] -  Check and adjust the misconfiguration.

L.406- "significantly increased" - Do not use terms that refer to results or numbers. Here is the conclusion section, answer your objectives, informing in your experimental conditions the best and/or the worst, and other relevant information for other researchers on the subject.

I think it would be important to include the limitations of the study in the paper. As a researcher, I read all of them before starting a new project. - "The major limitation of this research is the lack of commercially available AX preparation. Commercially available AX preparation could be used on a bigger scale and the research could be extended to for example other postponed baking methods or sensory analysis. We have added an appropriate information at the end of the discussion and in the conclusions."

Comments on the Quality of English Language

Minor editing of English language required

Author Response

The detailed responses to the reviewer's comments are provided in the file attached.
